# Pangenomic antiviral effect of REP 2139 in CRISPR/Cas9 engineered cell lines expressing hepatitis B virus surface antigen

Léna Angelo[1], Andrew Vaillant[2], Matthieu Blanchet[1,2]*, Patrick Labonté[1]*

**1** Institut National de la Recherche Scientifique–Centre Armand-Frappier Santé Biotechnologies, Laval, Canada, **2** Replicor Inc, Montréal, Canada

* patricklabonte@inrs.ca (PL); matthieublanchet@hotmail.com (MB)

**Data Availability Statement:** All relevant data are within the paper and its Supporting Information files.

## Abstract

Chronic hepatitis B remains a global health problem with 296 million people living with chronic HBV infection and being at risk of developing cirrhosis and hepatocellular carcinoma. Non-infectious subviral particles (SVP) are produced in large excess over infectious Dane particles in patients and are the major source of Hepatitis B surface antigen (HBsAg). They are thought to exhaust the immune system, and it is generally considered that functional cure requires the clearance of HBsAg from blood of patient. Nucleic acid polymers (NAPs) antiviral activity lead to the inhibition of HBsAg release, resulting in rapid clearance of HBsAg from circulation *in vivo*. However, their efficacy has only been demonstrated in limited genotypes in small scale clinical trials. HBV exists as nine main genotypes (A to I). In this study, the HBsAg ORFs from the most prevalent genotypes (A, B, C, D, E, G), which account for over 96% of human cases, were inserted into the AAVS1 safe-harbor of HepG2 cells using CRISPR/Cas9 knock-in. A cell line producing the D144A vaccine escape mutant was also engineered. The secretion of HBsAg was confirmed into these new genotype cell lines (GCLs) and the antiviral activity of the NAP REP 2139 was then assessed. The results demonstrate that REP 2139 exerts an antiviral effect in all genotypes and serotypes tested in this study, including the vaccine escape mutant, suggesting a pangenomic effect of the NAPs.

## Introduction

Despite an effective vaccine against hepatitis B virus (HBV), the World Health Organization (WHO) estimates that 296 million people are still living with chronic HBV infection, causing 820,000 deaths annually from cirrhosis and hepatocellular carcinoma [1]. During HBV infection, the most abundant circulating antigen is the hepatitis B surface antigen (HBsAg). HBsAg exists in three different isoforms which comprise the three HBV envelope proteins referred to as L-, M-, and S-HBsAg. The vast majority of these proteins assemble to form non-infectious subviral particles (SVP) which contain only trace amounts of L-HBsAg [2–4].

HBV can be divided into nine genotypes classified from A to I [5, 6] with an additional putative genotype J reported in one patient [7]. The worldwide distribution of genotypes among chronically HBV infected patients varies [8], yet over 96% of patients are infected by one

**Funding:** The work was co-funded by an Alliance grant from the Natural Sciences and Engineering Research Council of Canada and Replicor Inc. P.L. was the recipient of the Natural Sciences and Engineering Research Council -Alliance grant (558342-2020). L.A. was the recipient of a Canada Graduate Scholarships–Master's from the Natural Sciences and Engineering Research Council. Replicor Inc (represented by A.V.) provided support in the form of salaries for M.B., but did not have any additional role in the study design, data collection and analysis, decision to publish, or preparation of the manuscript. The specific roles of these authors are articulated in the 'author contributions' section.

**Competing interests:** I have read the journal's policy and the authors of this manuscript have the following competing interests: M.B. is an employee and a stakeholder of Replicor. A.V. is an employee and a shareholder in Replicor. This does not alter our adherence to PLOS ONE policies on sharing data and materials.

of the five most common HBV genotypes. Their proportions have been reported as follows: C being the most common with 26%, followed by genotype D (22%), E (18%), A (17%) and B (14%). Genotypes F to I together cause less than 2% of global chronic HBV infections [8].

All HBsAg isoforms contain the primary epitope of HBV required to induce protective antibody responses. This epitope is exposed on the surface of HBV virions and SVP particles and is called the "a" determinant [9]. This "a" determinant spans amino acid 124 to 147 [9, 10]. The appearance of the disease in a child who was vaccinated at birth following maternal transfer of HBV infection led to the discovery of vaccine escape mutations [11]. The first reported was a glycine (G) to arginine (R) substitution at position 145 within the "a" determinant, causing a conformational change [11, 12]. As HBV vaccines induce HBsAg antibodies against the "a" determinant region of HBsAg, some mutations occurring in this region can lead to vaccine escape [10]. Several variants were previously identified including D144A [13]. This variant has been reported to be found in genotypes A, C, D, E [10] and is included in this study. Patients infected with HBV vaccine escape mutants do not benefit from vaccination and are especially in need for effective treatments.

Importantly, HBsAg elimination from blood is considered as a hallmark of achieving functional cure [14] and cessation of therapy [15].

Nucleic acid polymers (NAPs) are amphipathic single-stranded phosphorothioated oligonucleotides [16]. The clinically active lead compound for NAPs is the REP 2139, a 40-mer with a $(AC)_{20}$ sequence, with full 2'O-methyl and 5'C methylation [17–20]. In monotherapy as well as in combination with immunotherapies such as pegylated interferons (pegIFN) and nucleos (t)ides analogues such as tenofovir disoproxil fumarate (TDF), the antiviral effect of REP 2139 against HBV leads to the inhibition of HBsAg release. This results in rapid clearance of HBsAg from circulation in phase IIA clinical trials in HBeAg negative and positive chronic HBV mono-infection and in HBV/HDV co-infection [18–20]. In the most recent phase IIA trial of NAP-based combination therapy, durable virological control of infection in the absence of therapy was achieved in 78% of participants, with 39% of participants further achieving functional cure [20]. We have previously succeeded in recapitulating the effect of NAPs *in vitro* [17, 21] in HepG.2.2.15 cells and were able to confirm the potent antiviral activity of REP 2139 on HBsAg secretion [21]. However, the efficacy of NAPs has only been demonstrated in a subset of genotypes [17–21]. Pangenomic effect assessment for any antiviral treatment is an important property to evaluate, since previous studies have demonstrated a genotype-dependent variability in the antiviral response to pegIFN in HBeAg +/- patients [22–24], and stronger pegIFN responses in genotypes presenting a lower HBsAg secretion [24–27]. Additionally, HBsAg turnover appears to vary with different HBV genotypes [28, 29].

In this study, REP 2139 antiviral effect is being assessed on HBsAg secretion from several HBV genotypes. HepG2 cells were engineered to secrete HBsAg from wildtype genotypes (A, B, C, D, E and G) and a vaccine escape mutant (D144A genotype D). These cell lines were created using the CRISPR/Cas9 technology to insert the open reading frame (ORF) of L, M and S-HBsAg and the HBx ORF into the safe-harbor AAVS1 located in chromosome 19. This locus has been reported to be a robust transgene expression site, reliable and safe for cell engineering, without detectable transcriptional perturbation of endogenous gene activity [30, 31]. Here, we report the pangenomic antiviral effect of REP 2139 on HBsAg secretion in our HepG2-derived genotype cell lines (GCLs).

## Materials and methods

### Phylogenic tree

GenBank accession numbers of sequences of L-HBsAg used in the phylogenic tree (Fig 1A) are listed in the legend. Evolutionary analyses were conducted in MEGA11 [32]. The

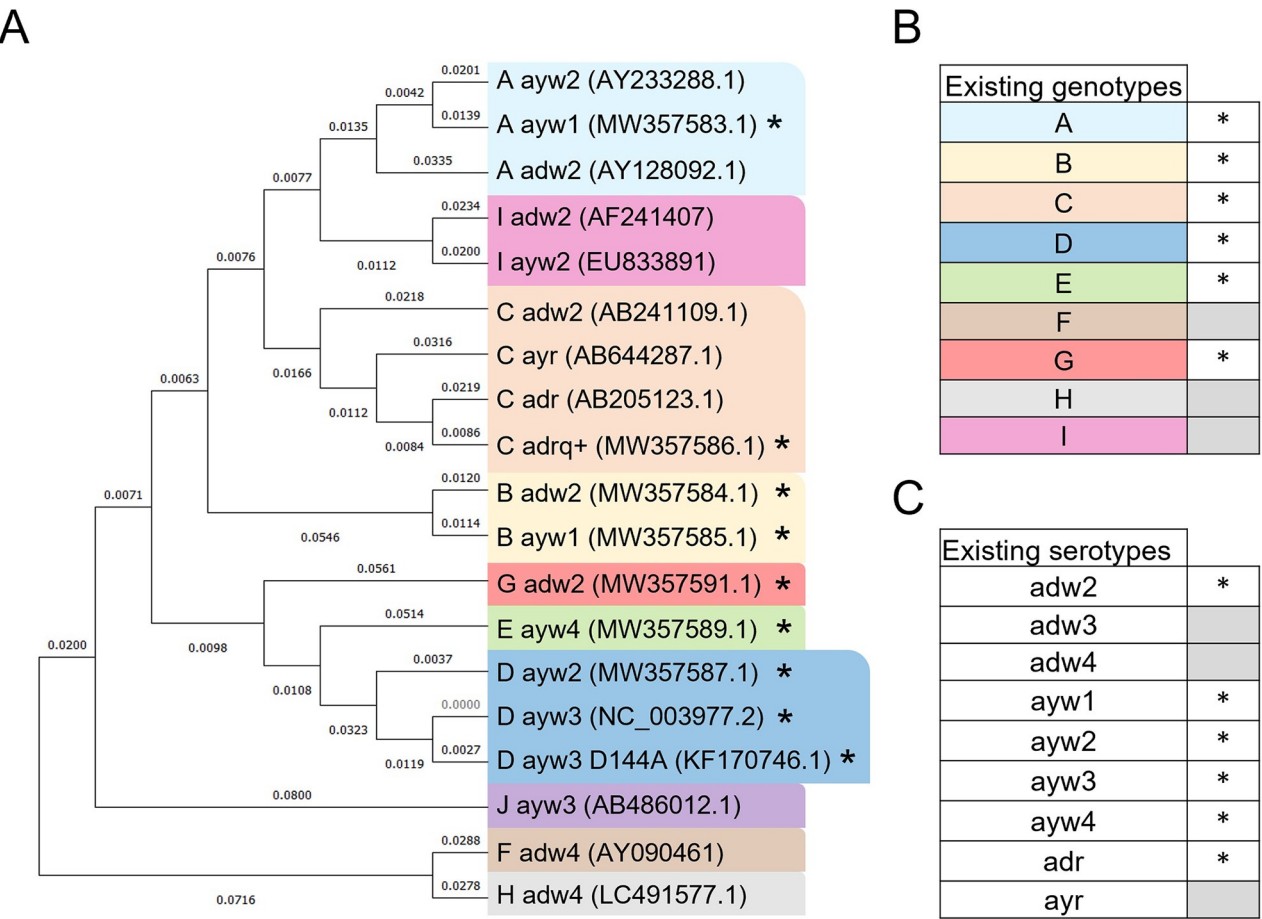

**Fig 1. Phylogenic classification of the L-HBsAg protein ORF according to genotypes and serotypes.** (A) Phylogenic tree for one of each genotype/ serotype couple existing for every genotype. GenBank accession numbers for the sequences included in the phylogenetic analyses are as follows: **AY233288.1**, **MW357583.1**, **AY128092.1, AF241407, EU833891**, **AB241109.1**, **AB644287.1**, **AB205123.1, MW357586.1, MW357584.1, MW357585.1**, **MW357591.1**, **MW357589.1**, **MW357587.1**, **NC_003977.2**, **KF170746.1, AB486012.1, AY090461**, and **LC491577.1**. (B) Existing HBV genotypes identified. (C) Existing HBV serotypes identified.(*) marks those evaluated in this study.

evolutionary history was inferred using the Neighbor-Joining method [33]. The evolutionary distances were computed using the Poisson correction method [34] and are in the units of the number of amino acid substitutions per site (next to the branches). This analysis involved 19 amino acid sequences corresponding to the entire L-HBsAg protein. All ambiguous positions were removed for each sequence pair (pairwise deletion option). There was a total of 540 positions in the final dataset.

## Cells and reagents

HepG2 cells were used for the construction of the genotype cell lines. HepG2.2.15 cells were used as a control in which NAP activity has been demonstrated [17, 21]. Cell lines were maintained in William's medium E (WME) complemented with 10% fetal bovine serum (FBS) and gentamicin. NAPs were prepared as previously described [35]. Treatment of cells was performed with stock solutions of REP 2139 in normal saline. The UNC7938 compound, a generous gift from Dr. Rudolph L. Juliano [36], was resuspended in DMSO. Puromycin dihydrochloride was purchased from Sigma-Aldrich (#P8833).

## Plasmids and cloning

Various pT7HB2.7 [37] plasmids containing PreS1, PreS2 and S genes from various genotypes and the X gene from genotype D ayw3 were graciously provided by Dr. Camille Sureau. Plasmids pAAVS1-HBsAg, pAAVS1-HBsAg-HBx forward and reverse were created from the plasmid AAVS1_Puro_Tet3G_3xFLAG_Twin_Strep (Addgene #92099) cleaved with SalI and NsiI with the subsequent cleaved fragment replaced with the HBsAg ORF +/- the HBx ORF from the pT7HB2.7. After confirmation of the HBsAg secretion from the pAAVS1-HBsAg-HBx forward construct (genotype D ayw3), we proceeded in the same way to construct similar plasmids for all genotypes. These plasmids were used as donors for the CRISPR/Cas9 knock-in to generate the GCLs. Cell lines generation for all GCLs is described in the CRISPR/Cas9 knock-in section.

Plasmid eSpCas9(1.1)_No_FLAG_AAVS1_T2 (Addgene #79888) was used for the Cas9 delivery.

## CRISPR/Cas9 knock-in

For each genotype, $5 \times 10^5$ HepG2 cells/well were seeded on collagen-coated 6-well plates and transfected the next day with Cas9/sgRNA coding plasmid and pAAVS1-puro-HBsAg-HBx (for each genotype) at a ratio of 1:1, using Lipofectamine™ 3000 Transfection Reagent (ThermoFisher Scientific) according to manufacturer's protocol. Cells were cultured for 14 days before addition of puromycin (0.5 μL/mL). When previously transfected cells were confluent, cells were trypsinized and plated at various densities ($2 \times 10^5$ and $1 \times 10^5$ cells/well in 6-well plates) and cultured with 0.5 μL/mL of puromycin for 52 days. Remaining living cells were trypsinized, pooled together as population cell lines, and HBsAg secretion in these cell lines was assessed before performing further experiments.

## Touch-down PCR

Genomic DNA (gDNA) was extracted using QIAamp® DNA Mini kit (Qiagen). Total cellular DNA concentrations were normalized following Nanodrop quantification by adjusting all gDNA concentration to 50 ng/μL. Amplification of gDNA was performed using iProof™ High-Fidelity DNA Polymerase (Bio-Rad) along with the following primers: forward primer (5′ – CCCTGGCCATTGTCACTTTG–3′) located in the chromosome 19 in 5' to the homology arm, and reverse primer (5′ –GAGTTCTTGCAGCTCGGTGAC–3′) located in the puromycin gene. PCR conditions were as follows: initial denaturation (3 min at 98°C) followed by 15 cycles of touch-down [denaturation: 10 s at 98°C; touch-down: 10 s from 72°C to 64.5°C (-0.5°C/cycle), elongation: 1 min at 72°C], then followed by 25 cycles of regular PCR (denaturation: 10 s at 98°C; annealing: 10 s at 64°C, elongation: 1 min at 72°C) and a final elongation (5 min at 72°C).

## Cell viability

Cell viability was assessed using total cellular protein concentration. Previously published data demonstrated that BCA assay provides similar assessment of cell viability compared to MTS assays in HepG2.2.15 cells [17]. Cells were lysed in Pierce lysis buffer (25 mM Tris-HCl pH 7.4, 150 mM NaCl, 1% NP-40, 1 mM EDTA, 5% glycerol) and the BCA assay was performed as per the manufacturer instructions (ThermoFisher Scientific; Pierce TM BCA Protein Assay Kit).

## ELISA

For HBsAg secretion assessment of the GCLs (Fig 4B), $7,5 \times 10^4$ cells/well were seeded on collagen-coated 24-well plates and cultured for 48 h. HBsAg secretion assessment following REP

2139 treatment is as indicated in Fig 5A. HBsAg quantification in supernatants was conducted using the GS HBsAg EIA 3.0 Kit (Bio-Rad) employing a standard curve from dilution of HepG2.2.15 supernatant. Presented results are normalized to total intracellular protein content (BCA). Means and standard deviations were calculated from replicate experiments (N = 3).

## Confocal fluorescence microscopy

$1 \times 10^5$ cells/well were cultured on collagen-coated glass coverslips and fixed the next day for 10 min in 4% paraformaldehyde. Cells were permeabilized for 30 min with 0.2% TritonX-100, followed by incubation with blocking solution (3% BSA, 10% FBS) for 1 h at RT, then labelled with an anti-HBsAg (1:150) from Abcam (ab9193) for 1 h at RT. Alexa Fluor® 488 AffiniPure Goat Anti-Horse IgG (H+L) (1:1000) (# 108-545-003) was incubated for 1 h at RT, followed by DAPI staining. Coverslips were then mounted on microscope slides using Prolong antifade reagent (ThermoFisher Scientific). Cells were analyzed using a confocal microscope (Zeiss LSM 780). Detector sensitivity was constant for all samples.

## Statistical analysis

Results shown represents the means of at least three independent experiments. Unpaired Student's t-test was performed for Fig 2D. One-way ANOVA analysis followed by a Dunnet's comparison test were performed in Fig 5B to identify statistically significant differences. P values below 0.05 were considered statistically significant (*, $P < 0.05$; **, $P < 0.01$; ***, $P < 0.001$; ****, $P < 0.0001$). All the statistical analysis were performed using Prism-GraphPad.

# Results

## Phylogenic classification of L-HBsAg proteins

A phylogenic classification of L-HBsAg for all the existing genotypes (A-I) associated with their existing serotypes is represented (Fig 1A). Based on a previous gathering of existing genotypes/serotypes [5], we selected one of each genotype/serotype couple to construct this phylogenic tree in order to have a clear and simple representation. Genotypes presented in this study are specified (Fig 1B), as well as serotypes (Fig 1C). According to this classification, our current study covers the majority of existing genotypes and serotypes found in chronic HBV infection worldwide.

## HPRE is essential for the secretion of HBsAg

Because the purpose of this study was to assess the REP 2139 antiviral effect on HBsAg secretion, we first aimed to create the simplest model for HBsAg secretion, expressing only HBV envelope proteins. To this end, a plasmid pAAVS1-HBsAg containing only the HBsAg ORF was constructed and transfected into HepG2 cells. At 48 h post-transfection, HBsAg secretion in the transfected cells was lower than HBsAg secretion in HepG2.2.15 cells (data not shown). Considering that a transient transfection where HBsAg is expressed from hundreds of copies of the plasmid, a stable cell line either homo- or heterozygous for the HBsAg gene would express even less HBsAg. Due to this observation, the importance and the necessity to include the HBV post-transcriptional regulatory element (HPRE) into our model was assessed.

The HPRE was previously reported to be important for the export of mRNA from the nucleus to the cytoplasm [38–40], thus being necessary for efficient translation and secretion of HBsAg. The HPRE sequence folds into an RNA secondary structure acting in a cis- and orientation-dependent manner [38–40]. Since all HBV mRNA share a common polyadenylation site, the HPRE is fully contained in PreCore/Core (C), L and M/S encoding mRNAs. This

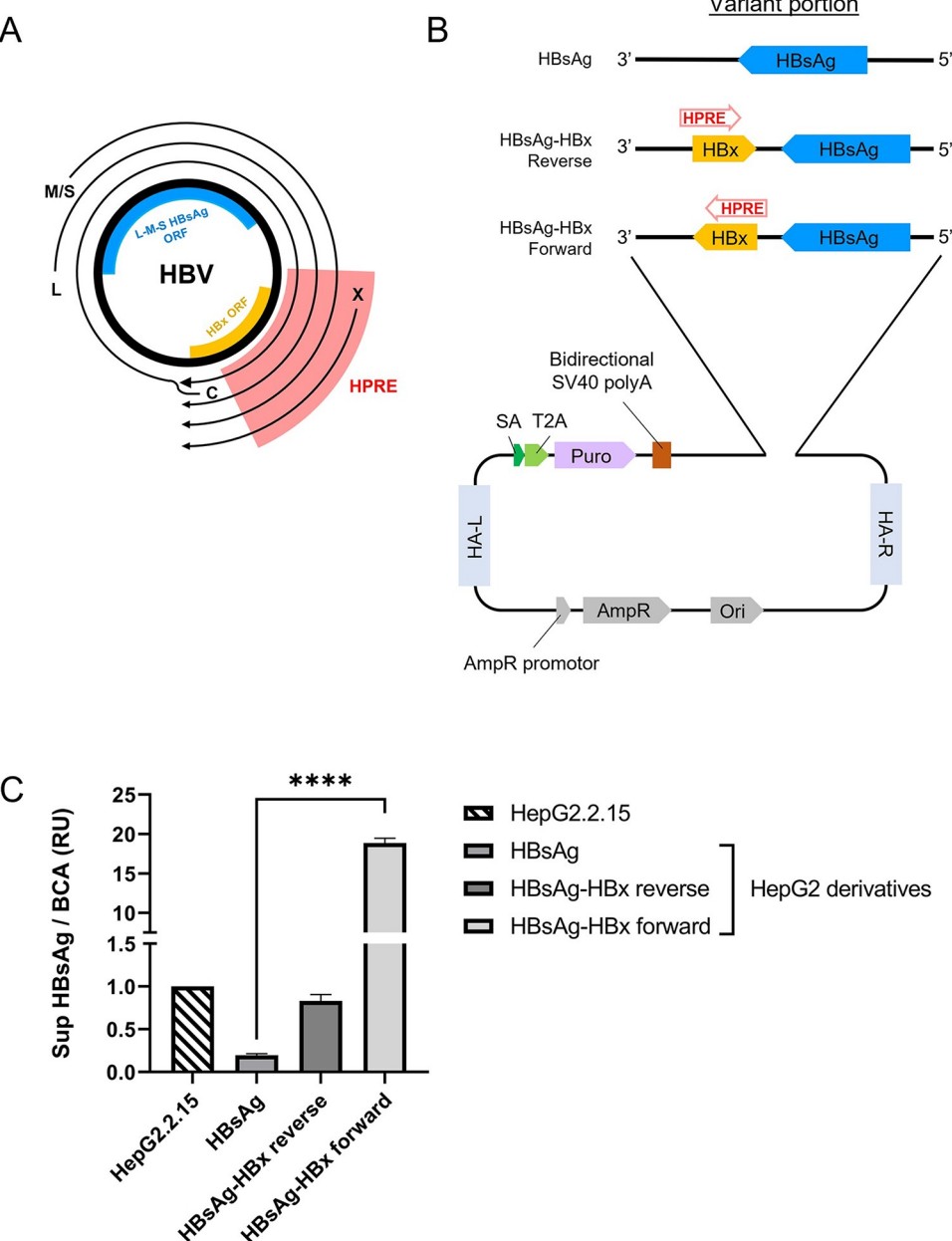

**Fig 2. Importance of the HBV post-transcriptional regulatory element (HPRE).** (A) Map of the four HBV mRNA showing S-HBsAg ORF and the localization of the HPRE. (B) pAAVS1-puro backbone used to assess the importance of the HPRE. The variant portion of the plasmid is described. HA-L/R, homology-arm left/right; Puro, puromycin resistance gene; SA, splice acceptor; T2A, *Thosea asigna* virus 2A peptide. (C) HBsAg secretion at 48 h post-transfection of the plasmids described above. Sup, supernatant; RU, relative unit. Unpaired Student's t-test was performed (****, P < 0.0001).

sequence is also partly contained in the HBx mRNA [41]. Thus, this sequence overlaps the ORF of the HBx protein (Fig 2A). To confirm the importance and the need for the HPRE in our model, two new plasmids were constructed: pAAVS1-HBsAg-HBx forward, containing the HBsAg with the HBx ORF in the proper orientation for the HPRE, and pAAVS1-HBsAg-HBx reverse, containing the HBx ORF in the reverse orientation (Fig 2B). These constructs

were transfected along the pAAVS1-HBsAg plasmid in HepG2 cells. Results confirm the previous observations from Huang and Liang [38] (Fig 2C). Indeed, cells transfected with the plasmid containing the HPRE in the proper orientation were able to increase the HBsAg secretion up to 96-fold in comparison with cells transfected with the plasmid expressing only HBsAg. Of note, the transfection with the plasmid bearing the reverse sequence of HPRE was able to induce a slight increase of the secretion of HBsAg, but to a much lower extent than with the proper HPRE orientation. Altogether, these results confirm the need of the HPRE and its orientation-dependent function in mRNA export.

Based on these results, the proper orientation of HBx ORF was conserved in the original HBsAg donor plasmids from which all genotypes were derived. All inserted HBV sequences contained HBsAg and HBx ORFs along with their own endogenous promoters (Fig 3A).

## HBsAg-HBx ORF is properly inserted into the AAVS1 safe-harbor

To confirm the proper CRISPR/Cas9 insertion of transgenes into the AAVS1 safe-harbor (Fig 3B), cellular DNA was extracted and amplified by touch-down PCR. For all genotype insertions analyzed, the expected DNA amplicon of 1241 bp was observed (Fig 3C), as well as the expected fragments of 1017 bp and 224 bp obtained after BglII digestion of the PCR product (Fig 3D). Altogether, these data confirm that all transgenes were properly inserted in the AAVS1 locus.

## Genotype cell lines express S-HBsAg

After insertion monitoring, we sought to evaluate the expression and secretion of HBsAg in the different GCLs. To this end, the intracellular HBsAg was first analyzed by indirect immunofluorescence (Fig 4A). Results showed that all cell lines expressed S-HBsAg with similar intracellular distributions. Cell cultured supernatants from the GCLs were harvested and analyzed for the secretion of HBsAg by ELISA (Fig 4B). Results showed that most GCLs secreted HBsAg at levels similar to HepG2.2.15 cells (D ayw3, B ayw1, B adw2, D ayw2, mutant D144A and E ayw4, G adw2). However, genotype A ayw1 and C adrq+ secreted 3 times more HBsAg protein than HepG2.2.15. Overall, all of the GCLs expressed and secreted HBsAg.

## REP 2139 is effective against all HBV genotypes

The *in vitro* antiviral effect of REP 2139 on HBsAg synthesis/secretion was previously demonstrated in HepG2.2.15 cells [17, 21]. The experimental design used to confirm the effect of REP 2139 in the GCLs and in the HepG2.2.15 control cells is described in Fig 5A. The results demonstrate that REP 2139 inhibits the secretion of HBsAg in every genotype tested in this study (Fig 5B), the statistical analyses of Fig 5B are presented in S1 Table, and the half maximal effective concentration (EC50) for each cell line is listed in Table 1. Importantly, we observed that the inhibition of HBsAg secretion after REP 2139 treatment is greater in the HepG2.2.15 cell than in all GCLs. Possible reason(s) explaining this discrepancy are explored in the discussion. Nevertheless, all EC50 are in the nanomolar range, which is in line with previous *in vitro* and clinical studies [17–20].

## Discussion

This study is the first to demonstrate the pangenomic antiviral effect of NAPs on HBsAg secretion. Indeed, although some genotypes are not represented, a large majority of current global chronic HBV infections (~ 96%) are covered. Genotypes F, H, I and J, which were not included in this study, represent less than 2% of global chronic HBV infections [8]. Also, since the

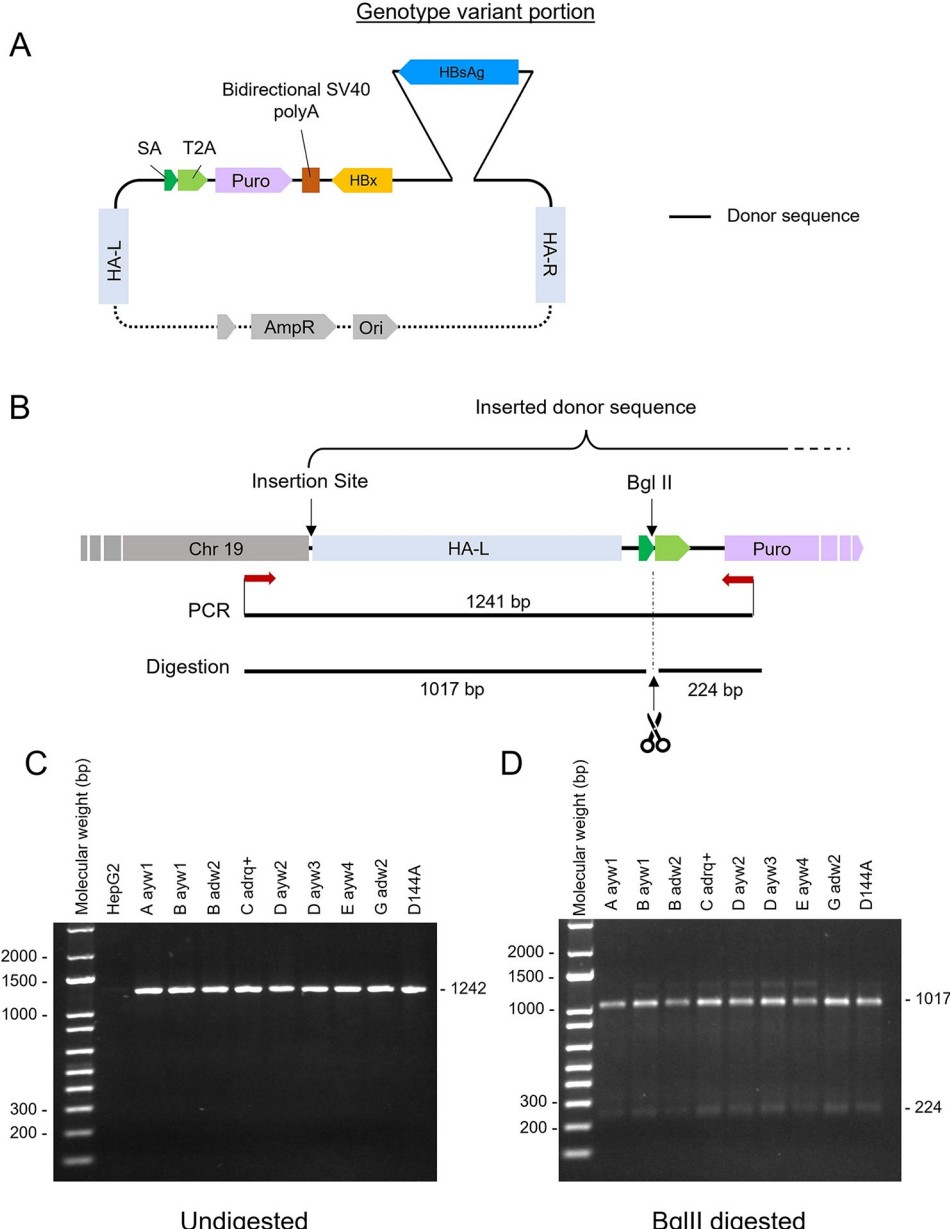

**Fig 3. Integration of HBsAg-HBx sequence into the AAVS1 safe-harbor.** (A) Map of the plasmid used for CRISPR/Cas9 insertion of HBsAg from variant genotypes and HBx. (B) Chromosome 19 (Chr 19) modified to the insertion site by insert of the transgene. The left homology-arm (HA-L) and the puromycin resistance gene from plasmid described in A are represented. Experimental design on this gDNA is as shown, primers attachment sites are indicated in red arrows, and the restriction site of BglII is also indicated. (C) Agarose gel presenting the PCR amplification product of the insertion site. (D) Agarose gel presenting the BglII digested PCR product.

L-HBsAg from genotype I is genetically similar to L-HBsAg from genotype A and C (Fig 1), and genotype J has been reported to be closely related to gibbon/orangutan genotypes and human genotype C [7], our results suggest that REP 2139 would be efficient in these genotypes as well due to the effect observed in genotypes A and C (Fig 5B). Importantly, the antiviral effect observed with the vaccine escape mutant D144A suggests that REP 2139 does not target

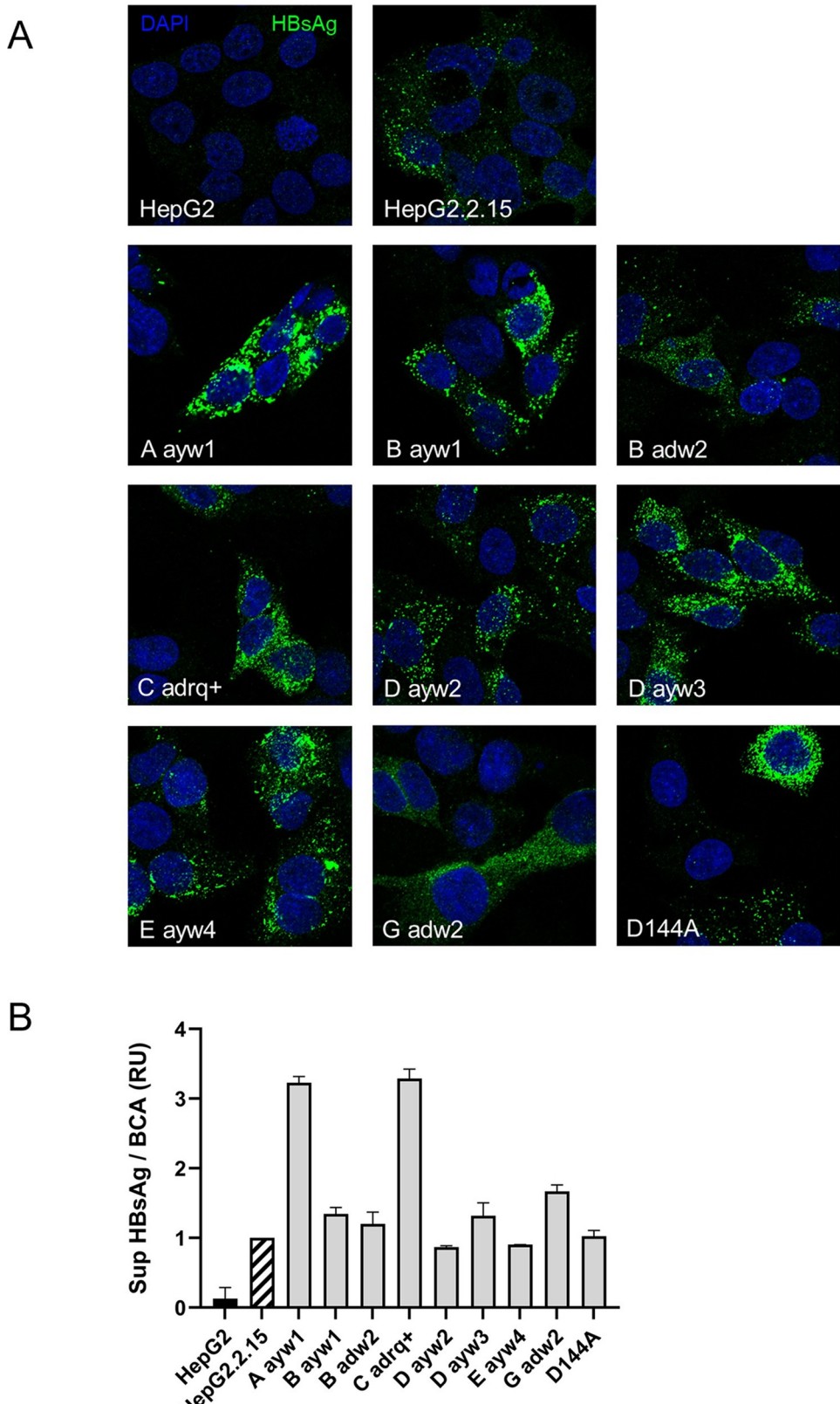

**Fig 4. Analysis of HBsAg expression and secretion from the indicated cell lines.** (A) Intracellular HBsAg expression analysis was conducted by confocal microscopy. Nucleus are stained in blue and HBsAg in green. (B) Secreted HBsAg from each cell line was assessed by ELISA, normalized to cell viability and to HepG2.2.15 secretion levels. Sup, supernatant; RU, relative unit.

a specific region in the "a" determinant. Altogether, these results show an antiviral effect on every tested genotypes, serotypes, and escape mutant and are consistent with the fact that NAPs do not directly target HBsAg [42] and indeed, does not depend on HBsAg sequence or conformation. Altogether our data strongly suggest a pangenomic antiviral effect of REP 2139 on HBsAg secretion.

GCLs were constructed using the CRISPR/Cas9 technology and the proper insertion of the transgene was confirmed by touch-down PCR (Fig 3). Transgene copies can be inserted either in a heterologous or in a homologous way. Indeed, the HepG2 parental cell line of the GCLs is diploid for the chromosome 19 [43] bearing the targeted AAVS1 locus. Based on this statement, one could expect the HBsAg secretion to be within a two-fold range between the various cell lines. Several factors could explain that those variations are up to 3-fold the lowest secreting cell line. Firstly, the antibody used to detect HBsAg in immunofluorescence (Fig 4A) or the ELISA kit used to detect HBsAg in the supernatant (Fig 4B) were the same for all genotypes in each assay, and no variation of antibody nor ELISA kit was employed. This could lead to variation in the relative detection depending on the specificity of the antibodies used in each technique to each genotype. Secondly, secretion of HBsAg has been reported to vary according to the genotype, with the most important release associated to genotype C [29]. In our hand, genotype C and A were clearly the most efficiently secreted (Fig 4B). Further investigations will be required to better understand the genotype-dependent HBsAg regulation of secretion.

Regarding the reduction of the amplitude effect of NAPs in the GCLs compared to HepG2.2.15 cells observed in Fig 5B, several factors could explain this variation. The first likely reason can be the difference of cell line: HepG2.2.15 and HepG2 are two distinct cell lines, and GCLs are derived from HepG2. The REP 2139 antiviral activity relies on the ability of the UNC7938 to release REP 2139 from endosomes [17, 36, 44, 45]. From previous experiments using the UNC7938 compound with REP 2139 in our lab, both cellular density and cell type strongly affected the efficacy of the UNC7938 to release the REP 2139 (data not shown). This observation could explain the difference observed in the antiviral amplitude effect between HepG2.2.15 and GCLs. The second factor explaining the amplitude difference could have been the genotypic variability. However, the genotype D ayw3 can be used as an anchor to link results between HepG2.2.15 and GCLs: this genotype is the HBV genotype of the HepG2.2.15 cell line and is also present in one of the GCLs. A comparison of the antiviral effect observed in HepG2.2.15 cells and the GCL genotype D ayw3 demonstrate that for a same genotype, the amplitude of the REP 2139 antiviral effect is different between both cell lines, excluding the genotypic variability as an explanation for the amplitude variation. Another factor that could explain this variation is the absence of the complete virus. Previous studies demonstrated that L-HBsAg can interfere with the secretion process [28], and an accumulation of L- and M-HBsAg can lead to endoplasmic reticulum stress [46]. As the S- and M-HBsAg are mainly used in the production of SVP and the L-HBsAg is essential for Dane particles assembly [47], which does not occur in our model due to the absence of other viral component, we suggest that the unused L-HBsAg could impact the secretion process in our GCLs, leading to a reduced amplitude of antiviral effect. While the EC50 are still in the nanomolar range, we believe that the *in vivo* effect of REP 2139 would be similar between all genotypes in large scale clinical trials, in accordance with previous results observed in the small scale clinical trials where NAPs were effective in genotypes D, C and A [18].

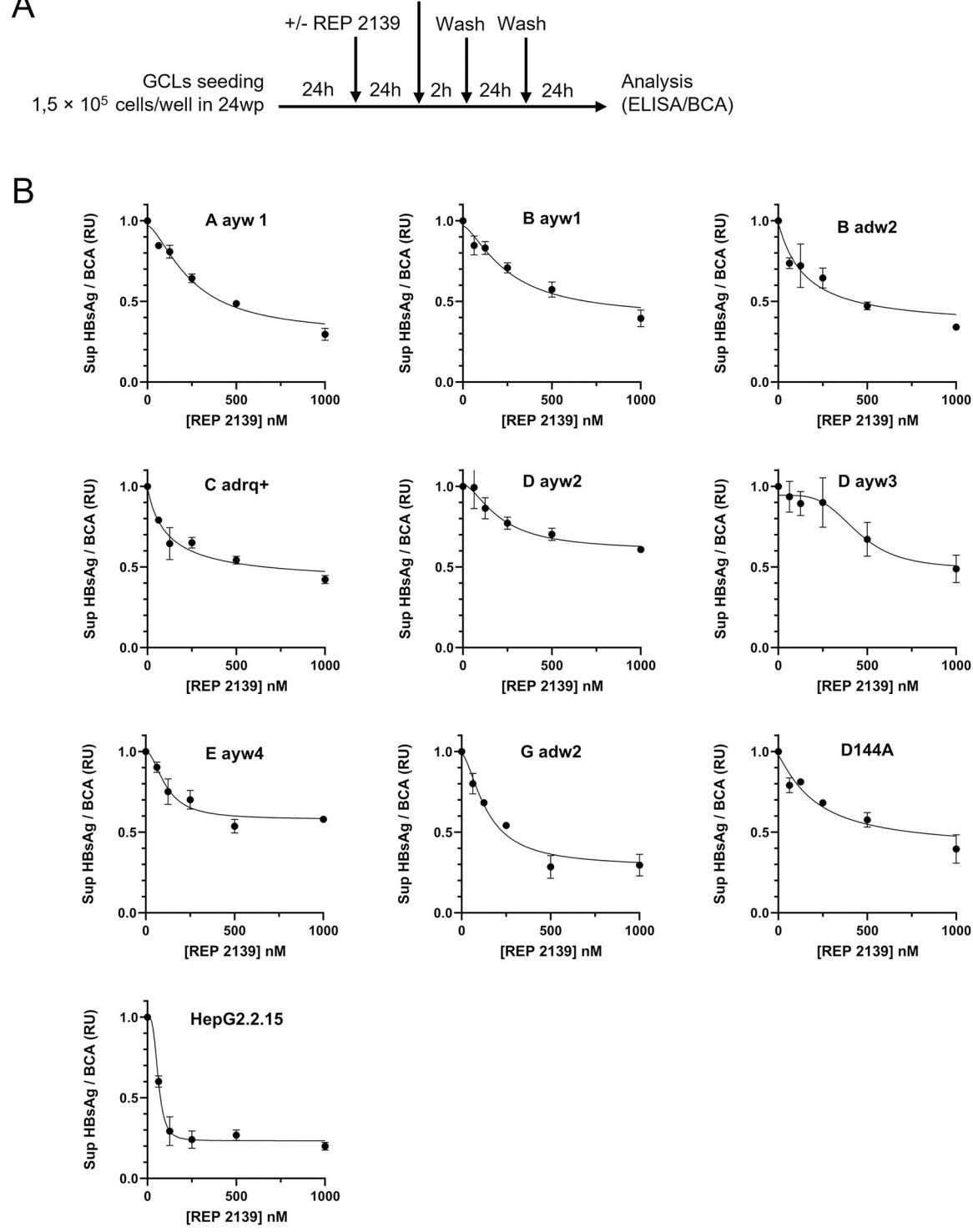

**Fig 5. Antiviral effect of REP 2139 in GCLs and HepG2.2.15 cells.** (A) Experimental design is as indicated. (B) Comparative antiviral effect of REP 2139 on HBsAg secretion for all indicated cell lines normalized to cell viability (BCA) and to 0 nM REP 2139. Sup, supernatant; RU, relative unit.

**Table 1. EC50 of REP 2139 in GCLs and HepG2.2.15 cells.** EC50 are defined by the concentration of REP 2139 needed to have half-maximum response. EC50 values were calculated using a non-linear regression curve in Prism-GraphPad.

| Gentoype/serotype | EC 50 (nM) |
|---|---|
| HepG2.2.15 | 61 |
| A ayw1 | 248 |
| B ayw1 | 255 |
| B adw2 | 149 |
| C adrq+ | 117 |
| D ayw2 | 207 |
| D ayw3 | 446 |
| E ayw4 | 113 |
| G adw2 | 142 |
| D144A | 211 |

## Conclusions

While the antiviral activity of NAPs has been already proven *in vivo* and *in vitro* on a limited number of genotypes, this study is the first to observe the pangenomic effect of REP 2139. CRISPR/Cas9 engineered cell lines expressing various HBV genotypes and serotypes allowed a stable and expression of HBsAg (cells cultured for more than 4 months) from HBsAg promoters, providing a robust and reliable model for the assessment of REP 2139 antiviral effect.

Importantly, these results also demonstrated an antiviral effect in a vaccine escape mutant, suggesting an effect in the other escape mutants. All together, these data strongly suggest that NAPs therapy could results in positive clinical outcomes regardless of genotype or serotype in chronically infected patients, as well as for patients infected with HBV escape mutants.

## Supporting information

**S1 Fig. Cell viability assessment of REP 2139 assay in GCLs.** Cell viability was assessed for the indicated cell lines for each REP 2139 concentration, by performing a BCA protein assay and measurement of the absorbance at 560nm. RU, relative unit.
(PDF)

**S1 Table. Statistical analyses results of REP 2139 antiviral effect.** One-way ANOVA analysis followed by a Dunnet's comparison between various REP 2139 concentration and 0nM REP 2139 were performed. (*, $P < 0.05$; **, $P < 0.01$; ***, $P < 0.001$; ****, $P < 0.0001$). Statistical analyses were performed using Prism-GraphPad.
(PDF)

**S1 Raw images. Raw images of agarose gel electrophoresis in Fig 3.**
(PDF)

## Acknowledgments

We thank the Dr. Camille Sureau for generously providing the pT7HB2.7 plasmids containing various HBsAg and HBx ORF used in this study. We thank the Dr. Rudolph L. Juliano for generously providing the UNC7938 compound. We are grateful to Jessy Tremblay for the technical assistance during imaging at confocal microscopy.

## Author Contributions

**Conceptualization:** Léna Angelo, Matthieu Blanchet, Patrick Labonté.

**Data curation:** Léna Angelo, Patrick Labonté.

**Formal analysis:** Léna Angelo, Matthieu Blanchet, Patrick Labonté.

**Funding acquisition:** Andrew Vaillant, Patrick Labonté.

**Investigation:** Léna Angelo.

**Methodology:** Léna Angelo, Matthieu Blanchet, Patrick Labonté.

**Supervision:** Matthieu Blanchet, Patrick Labonté.

**Validation:** Léna Angelo, Matthieu Blanchet.

**Visualization:** Léna Angelo.

**Writing – original draft:** Léna Angelo.

**Writing – review & editing:** Léna Angelo, Matthieu Blanchet, Patrick Labonté.

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
