## [Decision Letter · Decision Letter 0]

30 Jun 2023

PONE-D-23-18377Pangenomic antiviral effect of REP 2139 in CRISPR/Cas9 engineered cell lines expressing hepatitis B virus HBsAgPLOS ONE

Dear Dr. Labonté,

Thank you for submitting your manuscript to PLOS ONE. After careful consideration, we feel that it has merit but does not fully meet PLOS ONE’s publication criteria as it currently stands. Therefore, we invite you to submit a revised version of the manuscript that addresses the points raised during the review process. Both reviewers underlined modifications to perform in the text and better explanations or precisions to provide along the paper.

We look forward to receiving your revised manuscript.

Kind regards,

Isabelle Chemin, PhD

Academic Editor

PLOS ONE

Journal Requirements:

2. Please expand the acronym “NSERC” (as indicated in your financial disclosure) so that it states the name of your funders in full.

3. Thank you for stating the following in the Competing Interests/Financial Disclosure* (delete as necessary) section: 

   "I have read the journal's policy and the authors of this manuscript have the following competing interests:

     M.B. is an employee and a stakeholder of Replicor.

     A.V. is an employee and a shareholder in Replicor."

We note that one or more of the authors are employed by a commercial company: name of commercial company. 

Reviewers' comments:

Reviewer's Responses to Questions

**Comments to the Author**

1. Is the manuscript technically sound, and do the data support the conclusions?

Reviewer #1: No

Reviewer #2: Partly

2. Has the statistical analysis been performed appropriately and rigorously? 

Reviewer #1: N/A

Reviewer #2: No

3. Have the authors made all data underlying the findings in their manuscript fully available?

Reviewer #1: Yes

Reviewer #2: No

4. Is the manuscript presented in an intelligible fashion and written in standard English?

Reviewer #1: No

Reviewer #2: Yes

5. Review Comments to the Author

Reviewer #1: In this study, Angelo et al investigated the antiviral effect of REP2139 on HBsAg expression/secretion in various engineered HepG2 cell lines expressing HBsAg from different genotypes. Although previous studies have included genotypes A, C, and D, in this study the HBsAg from the most prevalent genotypes (A, B, C, D, E, G), which account for over 96% of human cases, were included. The authors found that REP 2139 exerts antiviral effects on all genotypes, as well as on the vaccine escape mutant. This study is neither comprehensive nor innovative. I have several major comments below to improve this manuscript:

Major comments:

1. The authors only showed the effects in the stable HepG2 cell lines. To further strengthen the conclusion from stable cell lines, the authors need to use infection systems (i.e., HepG2-NTCP and/or primary human hepatocytes) to confirm their finds.

2. In Fig 1, the phylogenetic tree of L-HBsAg is not sufficient. The authors need to include the phylogenetic analyses of HBV polymerase ORF and the full-length DNA sequence of these strains.

3. In Fig 2, it is not surprising that the authors’ construct expressing HBsAg would not support HBsAg secretion. Because there is not polyadenylation signal for HBsAg RNA transcription on this construct. Flipping of HBx sequence alone does not sufficient to conclude that HRPE is important as it also affected the polyA signal sequence. The authors need to dissect the element of polyA signal and HRPE individually from HBx coding sequence in this experiment.

4. REP2139 inhibits HBsAg SVP assembly and secretion, so it should not affect vaccine escape mutant D144A. In addition, the previous study already included some genotypes (A, C, and D). The only additional novel information from this study is genotypes B, E, and G. To improve the novelty of this study, investigations on mutants that affect the SVP assembly and secretion, eg., amino acid between 156 and 169 of S-HBsAg are suggested (Yang et al., 2021, J Virol, doi:10.1128/jvi.02399-20).

Minor point:

1. The sentence in page 3, line 44 “Importantly, HBsAg elimination from blood is required for achievement of functional cure [5] and cessation of therapy [6].” is controversial. It is impossible to eliminate HBsAg from the integrated HBV DNA. It might be better to change it into “Importantly, HBsAg elimination from blood is considered as the hallmark of achieving functional cure [5].”.

2. Page 3 Line 52-53, “In the most recent phase IIA trial of NAP-based combination therapy, durable immunological control of infection in the absence of therapy was achieved in 78% of participants, with 39% of participants further achieving functional cure [10]. How to define the immunological control? By seroconversion or by virological control? In ref 10, it states the detection of HBsAg below the LLOQ. Thus, the authors should use virological control here.

Reviewer #2: Overview:

The authors have shown successful generation of different genotype cell lines (GCLs) expressing hepatitis B surface antigen (HBsAg). This manuscript is the first to demonstrate the pangenomic antiviral effect of a nucleic acid polymer (NAP) such as REP 2139 against HBsAg expression. The results also demonstrated an antiviral effect in a vaccine escape mutant implying that NAPs could have an influence on escape mutants. However, the GCLs that were generated did not incorporate entire HBV genomes and could therefore not recapitulate natural viral replication. Although REP 2139 seems to have an antiviral effect on HBsAg from different HBV genotypes, this effect may be different during natural infection. In its current form, the manuscript is not suitable for publication. Points that need to be addressed are below.

Major points

Title:

1. The manuscript is titled ‘Pangenomic antiviral effect of REP 2139 in CRISPR/Cas9 engineered cell lines expressing hepatitis B virus HBsAg’. The title needs to be rephrased to something along the lines of ‘Pangenomic antiviral effect of REP 2139 in CRISPR/Cas9 engineered cell lines expressing hepatitis B virus surface antigen’. The title indicates that the cell lines replicated HBV, whereas it was only hepatitis B surface antigen (HBsAg) that was expressed.

Abstract and Introduction:

1. The significance of NAPs in clinical trials was mentioned, however the link between NAPs and how they target subviral particles (SVPs) was not described. Previously described mechanisms (found in references 7, 11 and 12) should be included in this manuscript.

2. Authors should describe clearly that the NAP used in the manuscript is REP 2139 and provide more background on this compound.

3. The structure of the abstract and introduction should be revised. Background information on HBV should include the different genotypes followed by the description of NAPs and then conclude with the importance of REP 2139 and the relevance of this NAP in the experiments conducted.

Materials and Methods

1. While the authors have included the relevant methods, several important details have not been described. Seeding and confluency of the GCLs as well as the HepG2 cells were not indicated for CRISPR/Cas9 knock-in, confocal fluorescence microscopy and ELISA experiments. The time points were also not listed for any of the confocal fluorescence microscopy and ELISA experiments. Overall, very little information is provided and these experiments will not be able to be repeated.

2. Statistical analyses were not performed throughout the manuscript and should be described in detail in the Materials and Methods section.

3. Lines 108-117: Plasmids and cloning. The authors need to elaborate further on how the cell lines were generated once pAAVS1-HBsAg-HBx forward were cloned (for the different genotypes)

4. Lines 126-127: Touch down PCR. The authors should specify how normalization was carried out.

5. Lines 136-141: Cell viability. The authors should clarify how the BCA would determine cell viability. This is unconventional and of questionable reliability.

Results

1. Figure 1: Phylogenic classification of L-HBsAg proteins should be placed in the supplementary section. Information provided in the figure is already well established.

2. Figure 2B: What was the significance of including the HBx ORF in the reverse orientation? The HPRE has already been shown to play an important role. The arrows annotating the HPRE orientation are also incorrectly represented. They should be reversed points towards the right for 5’ to 3’ forward orientation.

3. Figure 2C and Figure 4B: The y-axis has an abbreviation ‘Sup HBsAg (RU)’ which has not been fully described in the figure legend. Results from the HepG2.2 15 cells appear to be normalised. An explanation for this and statistical analyses should be included.

4. The authors indicate the insertion of the HBsAg-HBx ORF into the AAVS1 safe-harbour site in Figure 3 and only provide digestion restriction results to ascertain the PCR products. Sequencing should be performed to further validate the presence of genotype-specific sequences and their correct integration.

5. The authors should generate entire HBV-replicating cell lines to assess pangenomic effects more accurately.

6. Figure legends for Figures 3, 4 and 5 are not comprehensive enough.

7. BCA assay was stated to be performed in the methodology section (Line 136 to 141) to assess cell viability however data for this assay has not been provided.

8. Line 188: Provide reference.

9. Lines 236-237: ‘Importantly, we observed that the inhibition of HBsAg secretion after REP 2139 treatment is faster and greater in the HepG2.2.15 cell than in all GCLs’. Although greater inhibition of HBsAg secretion after REP 2139 treatment was observed in HepG2.2.15 cells (Figure 5), it is misleading to state that ‘faster’ inhibition was observed.

Discussion

1. Lines 266-267: This section covers important concepts and should be explained more thoroughly.

2. Lines 277-281: This explanation is not clear and should be rephrased.

Conclusion

1. Lines 296-297: Time course data has not been shown to verify stable expression of HBsAg from the GCLs.

Minor Points

1. Line 20: Mention SVP as non-infectious and Dane as infectious.

2. Line 21: Write out HBsAg in full, change “admitted” to “considered”.

3. Line 30: REP 2139 is not referred to as a NAP.

4. Line 31: Remove the word “in” “in the vaccine escape mutant”.

5. Line 39: Remove 2019.

6. Line 46: Discuss the structure of the NAPs in greater detail and link it to the SVPs.

7. Line 56: Rewrite “in a subset of genotypes including genotype D [8, 10-12], and genotypes C and A [8]” to “in a subset of genotypes including genotypes A, C and D [8-12].

8. Line 64: Remove “in different geographical regions.”

9. Line 73: Provide reference.

10. Line 87: As previously mentioned, provide context for REP 2139.

6. PLOS authors have the option to publish the peer review history of their article (what does this mean?). If published, this will include your full peer review and any attached files.

Reviewer #1: No

Reviewer #2: No

---

## [Author Response · Author response to Decision Letter 0]

8 Aug 2023

Reviewers' comments:

Reviewer's Responses to Questions

Reviewer #1: In this study, Angelo et al investigated the antiviral effect of REP2139 on HBsAg expression/secretion in various engineered HepG2 cell lines expressing HBsAg from different genotypes. Although previous studies have included genotypes A, C, and D, in this study the HBsAg from the most prevalent genotypes (A, B, C, D, E, G), which account for over 96% of human cases, were included. The authors found that REP 2139 exerts antiviral effects on all genotypes, as well as on the vaccine escape mutant. This study is neither comprehensive nor innovative. I have several major comments below to improve this manuscript:

Major comments:

1. The authors only showed the effects in the stable HepG2 cell lines. To further strengthen the conclusion from stable cell lines, the authors need to use infection systems (i.e., HepG2-NTCP and/or primary human hepatocytes) to confirm their finds.

We are not sure we fully understand the question. Indeed, in the present study we are solely expressing non infectious subviral particles (SVP). In this context an HepG2-NTCP infection model is irrelevant. The present study is meant to directly evaluate the capability of NAPs to inhibit HBsAg/SVP secretion in a simple and comparable fashion for various genotypes. The integration of the diverse HBsAg in the same chromosomal location (the AAVS1 safe harbour) is, to our view, ideal to analyse the effect of NAPs (here REP 2139) on HBsAg secretion.

2. In Fig 1, the phylogenetic tree of L-HbsAg is not sufficient. The authors need to include the phylogenetic analyses of HBV polymerase ORF and the full-length DNA sequence of these strains.

While we understand the reviewer comment we want to emphasize the fact that our focus is the variability of HBsAg sequences. In addition, the genotypes used in this study were provided by Dr Camille Sureau as derivatives of the pT7HB2.7 plasmid and thus do not include the whole genomic sequence. The important point here is to be able to clearly attribute the proper genotype/serotype to each sequence used. The phylogenetic tree used here thus focuses on the HBsAg sequences. 

3. In Fig 2, it is not surprising that the authors’ construct expressing HBsAg would not support HBsAg secretion. Because there is not polyadenylation signal for HBsAg RNA transcription on this construct. Flipping of HBx sequence alone does not sufficient to conclude that HRPE is important as it also affected the polyA signal sequence. The authors need to dissect the element of polyA signal and HRPE individually from HBx coding sequence in this experiment.

We are sorry for the confusion. There is a polyA signal in all constructs. The figure 2 has been modified to better detailed the constructs that have been used. At the beginning of this study, we wanted to use the simplest and minimal construct that would allow the stable expression of HBsAg. This is why we compared HBsAg expression with and without HBV HPRE. A clear depiction of our constructs is presented in Fig 2B.

4. REP2139 inhibits HBsAg SVP assembly and secretion, so it should not affect vaccine escape mutant D144A.

We don’t understand the reviewer’s comment. There is no obvious reason why the D144A mutant should not be affected? NAPs most likely affect the folding of HBsAg and D144A still need to be folded. D144A is a vaccine escape mutant because it fails to properly activate the immune system. From our perspective, it was very important to demonstrate that most genotype and vaccine escape mutants could be targeted by REP 2139. This antiviral is a new molecule that still needs to be studied in more details in regard to the encouraging clinical results.

In addition, the previous study already included some genotypes (A, C, and D). 

It’s true that some indication from patients or in the HepG2.2.15 have been reported and suggest that HBsAg from genotype A, C and D are sensitive to REP 2139. Nevertheless, a side-by-side study that include most genotypes as well as vaccine-escape mutant was, to our appreciation, of significant interest. 

The only additional novel information from this study is genotypes B, E, and G. To improve the novelty of this study, investigations on mutants that affect the SVP assembly and secretion, eg., amino acid between 156 and 169 of S-HBsAg are suggested (Yang et al., 2021, J Virol, doi:10.1128/jvi.02399-20). 

Here, we believe that it is not possible or very difficult for us to answer the reviewer comment. Indeed, it is implicitly irrelevant to evaluate HBsAg secretion in mutants for which intrinsic stability is affected. 

Minor point:

1. The sentence in page 3, line 44 “Importantly, HBsAg elimination from blood is required for achievement of functional cure [5] and cessation of therapy [6].” is controversial. It is impossible to eliminate HBsAg from the integrated HBV DNA. It might be better to change it into “Importantly, HBsAg elimination from blood is considered as the hallmark of achieving functional cure [5].”. 

The sentence has been modified accordingly (see line 66).

2. Page 3 Line 52-53, “In the most recent phase IIA trial of NAP-based combination therapy, durable immunological control of infection in the absence of therapy was achieved in 78% of participants, with 39% of participants further achieving functional cure [10]. How to define the immunological control? By seroconversion or by virological control? In ref 10, it states the detection of HBsAg below the LLOQ. Thus, the authors should use virological control here.

The sentence has been modified according to the reviewer comment (see line 76).

Reviewer #2: 

Overview:

The authors have shown successful generation of different genotype cell lines (GCLs) expressing hepatitis B surface antigen (HBsAg). This manuscript is the first to demonstrate the pangenomic antiviral effect of a nucleic acid polymer (NAP) such as REP 2139 against HBsAg expression. The results also demonstrated an antiviral effect in a vaccine escape mutant implying that NAPs could have an influence on escape mutants. 

However, the GCLs that were generated did not incorporate entire HBV genomes and could therefore not recapitulate natural viral replication. Although REP 2139 seems to have an antiviral effect on HBsAg from different HBV genotypes, this effect may be different during natural infection. In its current form, the manuscript is not suitable for publication. Points that need to be addressed are below.

The reviewer is right that our model does not recapitulate the full viral replication of those genotypes. However, the main goal of our study was to analyze and compare specifically the effect of REP 2139 on HBsAg secretion from various strains. Our results seem to confirm clinical results obtained with REP 2139 suggesting that our model is meaningful to analyze HBsAg secretion.

Major points

Title:

1. The manuscript is titled ‘Pangenomic antiviral effect of REP 2139 in CRISPR/Cas9 engineered cell lines expressing hepatitis B virus HBsAg’. The title needs to be rephrased to something along the lines of ‘Pangenomic antiviral effect of REP 2139 in CRISPR/Cas9 engineered cell lines expressing hepatitis B virus surface antigen’. The title indicates that the cell lines replicated HBV, whereas it was only hepatitis B surface antigen (HBsAg) that was expressed.

The title has been modified accordingly to the reviewer comment.

Abstract and Introduction:

1. The significance of NAPs in clinical trials was mentioned, however the link between NAPs and how they target subviral particles (SVPs) was not described. Previously described mechanisms (found in references 7, 11 and 12) should be included in this manuscript.

The reviewer is making a reference to our previous published works on the effect of NAPs, such as REP 2139 on HBV replication. Our main finding was that NAPs can efficiently inhibit HBsAg secretion (fully mentioned in the paper). However, so far, the mechanism by which REP 2139 inhibits HBsAg secretion remains to be elucidated. Our group is actively working on that part and should publish in a near future. 

2. Authors should describe clearly that the NAP used in the manuscript is REP 2139 and provide more background on this compound

The introduction has been complemented to reflect the reviewer comment. See line 69 “The clinically active lead compound for NAPs is the REP 2139, a 40-mer with a (AC)20 sequence, with full 2’O-methyl and 5’C methylation [17-20].”

3. The structure of the abstract and introduction should be revised. Background information on HBV should include the different genotypes followed by the description of NAPs and then conclude with the importance of REP 2139 and the relevance of this NAP in the experiments conducted.

We agree with the reviewer comment and the introduction was modified accordingly. See lines 48-86.

Materials and Methods

1. While the authors have included the relevant methods, several important details have not been described. Seeding and confluency of the GCLs as well as the HepG2 cells were not indicated for CRISPR/Cas9 knock-in, confocal fluorescence microscopy and ELISA experiments. The time points were also not listed for any of the confocal fluorescence microscopy and ELISA experiments. Overall, very little information is provided and these experiments will not be able to be repeated. 

We agree with the reviewer’s comment and additional information have been added throughout the Materials and Methods.

2. Statistical analyses were not performed throughout the manuscript and should be described in detail in the Materials and Methods section.

A section « Statistical analysis » has been added to the Materials and Methods section. We also included a table in supplementary to show the significance of the Fig. 5 data.

3. Lines 108-117: Plasmids and cloning. The authors need to elaborate further on how the cell lines were generated once pAAVS1-HBsAg-HBx forward were cloned (for the different genotypes)

A drawing was added in Fig 3 to better illustrated the cloning strategy and the section Plasmids and cloning was modified accordingly. See lines 113-123.

4. Lines 126-127: Touch down PCR. The authors should specify how normalization was carried out.

We agree. The sentence (line 135) was modified to: “Total cellular DNA concentrations were normalized following Nanodrop quantification by adjusting all gDNA concentration to 50 ng/µL.”

5. Lines 136-141: Cell viability. The authors should clarify how the BCA would determine cell viability. This is unconventional and of questionable reliability.

We understand the reviewer’s concern, but we previously compare normalization using BCA (protein quantity) and MTS (mitochondrial activity) for NAPs antiviral evaluation and we found that both methods were comparable. See Boulon et al., Antiviral Research 2020 DOI: 10.1016/j.antiviral.2020.104853

Results

1. Figure 1: Phylogenic classification of L-HBsAg proteins should be placed in the supplementary section. Information provided in the figure is already well established.

Although we understand the reviewer’s comment, the intent of the phylogenic classification is to allow the reader to easily visualize the distribution of the genotypes used in this study. Since PlosOne is an open access journal, we would like that Fig 1 remain in the core of the manuscript.

2. Figure 2B: What was the significance of including the HBx ORF in the reverse orientation? The HPRE has already been shown to play an important role. The arrows annotating the HPRE orientation are also incorrectly represented. They should be reversed points towards the right for 5’ to 3’ forward orientation.

During the course of this study, we started by NOT adding the HPRE hoping to get enough HBsAg to study the genotype in a minimal setting. However, it become evident that, using the HBV endogenous promoter, the HPRE was required. Finally, we do not understand the reviewer comment on the arrow orientation. To our understanding they are represented correctly. In any case, the new version of the Fig 2B should clarified the misunderstanding regarding the arrow orientation. A paragraph has also been added in order to clarify these observations (see lines 193-201). The HBx ORF in the reverse orientation was used as a control and make it easier to properly identify the contribution in cis of the HPRE. 

3. Figure 2C and Figure 4B: The y-axis has an abbreviation ‘Sup HBsAg (RU)’ which has not been fully described in the figure legend. Results from the HepG2.2 15 cells appear to be normalised. An explanation for this and statistical analyses should be included.

We agree and apologize for that mistake. The Y-axis have been changed to Sup HBsAg/BCA to reflect the fact that all sample were normalized on the BCA. Also, the statistical analysis was performed and added to the figures as requested (see Fig 2C and supplementary Fig 3)

4. The authors indicate the insertion of the HBsAg-HBx ORF into the AAVS1 safe-harbour site in Figure 3 and only provide digestion restriction results to ascertain the PCR products. Sequencing should be performed to further validate the presence of genotype-specific sequences and their correct integration. 

Fig 3 aim at making sure that the HBsAg gene integration was indeed in the AAVS1 site of chromosome 19. The fact that HBsAg protein was expressed is a confirmation of the protein integrity. Actually, it would have been difficult to sequence the whole chromosomic integrated fragment since we were working with cell populations rather than with pure clones. 

5. The authors should generate entire HBV-replicating cell lines to assess pangenomic effects more accurately.

We previously demonstrated that the main target of REP 2139 is HBsAg secretion. By using our setting, we can segregate the effect of NAPs of HBsAg secretion from others potential side effect. This was the intend of the research presented here. Evaluating the pangenomic antiviral effect of NAPs on complete HBV-replicating cell lines would present important technical challenges, such as delivering a very large cargo into the chromosome.

6. Figure legends for Figures 3, 4 and 5 are not comprehensive enough. 

We agree. Figure legends have been modified accordingly.

7. BCA assay was stated to be performed in the methodology section (Line 136 to 141) to assess cell viability however data for this assay has not been provided. 

We are now providing the BCA values in a new supplementary figure (see Fig S2)

8. Line 188: Provide reference. 

The references of Huang and Liang has been added (see line 205)

9. Lines 236-237: ‘Importantly, we observed that the inhibition of HBsAg secretion after REP 2139 treatment is faster and greater in the HepG2.2.15 cell than in all GCLs’. Although greater inhibition of HBsAg secretion after REP 2139 treatment was observed in HepG2.2.15 cells (Figure 5), it is misleading to state that ‘faster’ inhibition was observed. 

We agree and the new sentence is now: Importantly, we observed that the inhibition of HBsAg secretion after REP 2139 treatment is greater in the HepG2.2.15 cell than in all GCLs. Possible reason(s) explaining this discrepancy are explored in the discussion. (See lines 268-270)

Discussion

1. Lines 266-267: This section covers important concepts and should be explained more thoroughly. 

A new sentence was added to better explain the point: “Transgene copies can be inserted either in a heterologous or in a homologous way. Indeed, the HepG2 parental cell line of the GCLs is diploid for the chromosome 19 [43] bearing the targeted AAVS1 locus.” (See lines 297-299)

2. Lines 277-281: This explanation is not clear and should be rephrased.

We agree and a complete reformulation has been made (See line 310-324)

Conclusion

1. Lines 296-297: Time course data has not been shown to verify stable expression of HBsAg from the GCLs. 

The sentence has been modified to indicate that HBsAg expression was stable over a 4-month period.

Minor Points

1. Line 20: Mention SVP as non-infectious and Dane as infectious. Done

2. Line 21: Write out HBsAg in full, change “admitted” to “considered”. Done

 3. Line 30: REP 2139 is not referred to as a NAP.

4. Line 31: Remove the word “in” “in the vaccine escape mutant”. Done 

5. Line 39: Remove 2019. Done 

6. Line 46: Discuss the structure of the NAPs in greater detail and link it to the SVPs. Done

7. Line 56: Rewrite “in a subset of genotypes including genotype D [8, 10-12], and genotypes C and A [8]” to “in a subset of genotypes including genotypes A, C and D [8-12]. Done

8. Line 64: Remove “in different geographical regions.” Done 

9. Line 73: Provide reference. Done 

10. Line 87: As previously mentioned, provide context for REP 2139. Done

---

## [Decision Letter · Decision Letter 1]

9 Oct 2023

Pangenomic antiviral effect of REP 2139 in CRISPR/Cas9 engineered cell lines expressing hepatitis B virus surface antigen.

PONE-D-23-18377R1

Dear Dr. Labonté,

We’re pleased to inform you that your manuscript has been judged scientifically suitable for publication and will be formally accepted for publication once it meets all outstanding technical requirements.

Kind regards,

Isabelle Chemin, PhD

Academic Editor

PLOS ONE

Additional Editor Comments (optional):

Reviewers' comments:

Reviewer's Responses to Questions

**Comments to the Author**

1. If the authors have adequately addressed your comments raised in a previous round of review and you feel that this manuscript is now acceptable for publication, you may indicate that here to bypass the “Comments to the Author” section, enter your conflict of interest statement in the “Confidential to Editor” section, and submit your "Accept" recommendation.

Reviewer #1: All comments have been addressed

Reviewer #2: (No Response)

2. Is the manuscript technically sound, and do the data support the conclusions?

Reviewer #1: Yes

Reviewer #2: Yes

3. Has the statistical analysis been performed appropriately and rigorously? 

Reviewer #1: Yes

Reviewer #2: No

4. Have the authors made all data underlying the findings in their manuscript fully available?

Reviewer #1: Yes

Reviewer #2: Yes

5. Is the manuscript presented in an intelligible fashion and written in standard English?

Reviewer #1: Yes

Reviewer #2: Yes

6. Review Comments to the Author

Reviewer #1: The authors have appropriately addressed the concerns. Although with low novelty of this study, the story is complete and suitable for publication in PLoS One.

Reviewer #2: The manuscript is improved as a result of the revisions. Minor changes to finalise are the following:

1. LA description of how NAPs function is still missing. If mechanisms are not fully understood, this should be stated.

2. Some Supp. Figures do not seem to be cited in the body of the MS.

3. An explanation for why S Ag expression is adequate instead of complete replication of the virus should be included in the text.

4. Stats are missing from Fig. 4.

5. The conclusion states that stable expression for more than 4 months was achieved. However, the MS does not provide the relevant data. (This sentence starting on line 338 also needs editing.)

6. Methodology used to generate the data described in Fig. 5 seems to be missing.

Reverse orientation of sequences depicted in Fig. 2B is unconventional.

Line 33 ‘into’ needs to be replaced with ‘in’

Line 66-67 should be added to the previous paragraph.

Line 116, ‘REP 2139 antiviral effect is’ change to REP 2139 antiviral effect was’

7. PLOS authors have the option to publish the peer review history of their article (what does this mean?). If published, this will include your full peer review and any attached files.

Reviewer #1: No

Reviewer #2: No

---

## [Editor Report · Acceptance letter]

23 Oct 2023

PONE-D-23-18377R1 

Pangenomic antiviral effect of REP 2139 in CRISPR/Cas9 engineered cell lines expressing hepatitis B virus surface antigen. 

Dear Dr. Labonté:

I'm pleased to inform you that your manuscript has been deemed suitable for publication in PLOS ONE. Congratulations! Your manuscript is now with our production department. 

Kind regards, 

on behalf of

Mrs Isabelle Chemin 

Academic Editor

PLOS ONE